# Neuromyotonia with Central Nervous System Lesions following Quadrivalent Human Papilloma Virus Vaccination

**DOI:** 10.3390/vaccines10071132

**Published:** 2022-07-16

**Authors:** Maryam Hatami, Moritz Förster, Vivien Weyers, Saskia Räuber, Sven G. Meuth, David Kremer

**Affiliations:** Department of Neurology, University Hospital Düsseldorf, Medical Faculty, Heinrich Heine University, 40225 Düsseldorf, Germany; maryam.hatami@med.uni-duesseldorf.de (M.H.); moritz.foerster@med.uni-duesseldorf.de (M.F.); vivienstefanie.weyers@med.uni-duesseldorf.de (V.W.); saskiajanina.raeuber@med.uni-duesseldorf.de (S.R.); svenguenther.meuth@med.uni-duesseldorf.de (S.G.M.)

**Keywords:** neuromyotonia, human papilloma virus, quadrivalent vaccine, central nervous system lesions

## Abstract

Neuromyotonia is a rare peripheral nerve hyperexcitability syndrome often associated with antibodies directed against contactin-associated protein-like 2 and leucine-rich, glioma inactivated 1. The quadrivalent human papilloma virus vaccine Gardasil^®^, first approved in 2006, is known to be a highly effective prophylaxis against papillomavirus types 6, 11, 16, and 18. Molecularly, this non-infectious recombinant vaccine is based on purified L1 proteins from the human papilloma virus capsid. Since the approval of this vaccine, several studies have investigated its safety regarding the occurrence of autoimmune conditions following application. Here, we present the first case of neuromyotonia with active Gadolinium enhancing demyelinating central nervous system lesions following vaccination with Gardasil^®^.

## 1. Introduction

Neuromyotonia (NMT; *Isaacs syndrome*) is a rare predominantly acquired peripheral nerve hyperexcitability syndrome first described in 1961 by South African physician and physiologist Hyam Isaacs [1]. It features spontaneous and continuous muscle activity caused by repetitive motor unit action potentials in both active and resting states, resulting in delayed muscle relaxation. The exact cause of NMT is still unknown. However, thirty years after its first description, Sinha and colleagues suggested an underlying autoimmune mechanism and the efficacy of immunosuppressive treatment [2]. Since then, it has been shown that, indeed, a significant percentage of NMT cases have detectable autoantibodies, of which approximately 25% are associated with neoplasia [3]. The target antigens are currently considered to be contactin-associated protein-like 2 (CASPR2) and leucine-rich glioma-inactivated 1 (LGI1) on the presynaptic membrane of the neuromuscular junction [4], even though a lack of autoantibodies does not exclude the diagnosis. As a result, in some acquired cases patients may benefit from plasma exchange or intravenous immunoglobulin (IVIG). Conventionally, the symptoms in milder cases could be alleviated by membrane-stabilizing drugs such as valproic acid. For NMT occurring in conjunction with central nervous system (CNS) symptoms, the term *Morvan’s syndrome* has been established [5]. However, NMT *per se* is not associated with CNS manifestations. 

The quadrivalent human papilloma virus (qHPV) vaccine Gardasil®, first approved in 2006, is a highly effective prophylaxis against papillomavirus type 6, 11, 16, and 18, which predispose to cervical and anal cancers. Molecularly, this non-infectious recombinant vaccine is based on purified L1 proteins, which are virus-like particles (VLPs) of the HPV capsid. Since the approval of this vaccine, several studies have investigated its safety regarding the occurrence of autoimmune conditions following application. Here, we describe the first case of acquired NMT accompanied by asymptomatic demyelinating CNS lesions following the administration of the qHPV vaccine Gardasil^®^. 

## 2. Case Report

In April 2021, a 25-year-old Caucasian woman presented to our neurological outpatient clinic to obtain a second opinion regarding symptoms comprising joint deformities, stiffness of all limbs, and gait impairment. They had first occurred in November 2012, approximately three weeks after the patient had received the first dose of the quadrivalent human papilloma virus (qHPV) vaccine Gardasil^®^. Up until then, she had been a healthy athletic teenager with normal development and an unremarkable past medical history. The first abnormalities she noticed were fatigue, myalgia, and arthralgia, as well as erythema and edema of her small joints. These symptoms then partially remitted, but ultimately recurred when she received the second dose of Gardasil^®^ in December 2012, which was in line with the current European Medicines Agency (EMA) recommendations. Within a few months, she also developed persistent muscle cramps, twitching and fasciculations, hyperhidrosis, bilateral swan-neck deformities of her fingers (Figure 1A), and bilateral pes cavus (Figure 1B), which both persisted and significantly impaired her dexterity, gait, and overall quality of life. 

In 2013, a neurological work-up including MR-imaging was performed (Figure 2). Here, demyelinating juxtacortical (Figure 2A), periventricular (Figure 2B), and spinal (Figure 2D) lesions were found, of which two (precentral gyrus and spinal cord at level of Th10) were active as evidenced by Gadolinium (Gad) enhancement (Figure 2D). Cerebrospinal fluid (CSF) analysis showed a mild pleocytosis of 8 leukocytes/µL, normal protein, and positive oligoclonal bands (OCBs; pattern II). All analysed anti-neuronal and paraneoplastic antibodies in the CSF were negative, including anti-Hu, anti-Ri, anti-Yo, anti-Tr, anti-MAG, anti-myelin, anti-Ma/Ta, anti-GAD, anti-aquaporin-4, anti-glutamate (type NMDA), anti-glutamate (type AMPA), anti-GABA-B-Receptor, anti-Glycine-receptor, anti-amphiphysin, anti-CV2, anti-PNMA2 (Ma2/Ta), anti-recoverin, anti-SOX1, and anti-Titin. In addition, there was no indication of anti-LGI1- and anti-CASPR2-antibodies. Whole body MR-imaging yielded no signs of an underlying malignancy. In a rheumatological screening, the only finding was an isolated unspecific borderline-positive ANA titer of 1:80. Based on this workup, the then 17-year-old patient was diagnosed with a “multiple sclerosis (MS)-like” disorder. Steroid therapy was recommended, but her parents wished to obtain a second opinion before therapy initiation. Accordingly, between 2013 and April 2021 the patient consulted several other neurologists, rheumatologists, and internists, but did not receive a definite diagnosis. She had undergone numerous follow-up MRIs, in which the demyelinating lesions had decreased in size and number (imaging unavailable). She had not received any immunosuppressive therapy but, through intensive physiotherapy, was now able to walk unassistedly and climb a few stairs. However, running remained impossible. In addition to the above-mentioned deformities (i.e., pes cavus and swan-neck deformities), physical examination revealed an increased muscle tone of all limbs, myokymia, and fasciculations as well as myotonia, which was mostly pronounced when closing and opening her fists. The deep tendon reflexes were normal and there were no Babinski signs. In June 2021, the patient was admitted to our clinic for a thorough re-evaluation. New MRIs showed an increased lesion load in comparison to 2013 with new periventricular (Figure 2E), juxtacortical (Figure 2F), and spinal (Figure 2G) lesions, but showed no Gad enhancement.

Motor nerve conduction studies showed a borderline-increased distal motor latency (DML) of 4.22 ms (normal range <4.20 ms) and a pathological nerve conduction velocity of 37.8 m/s (normal range > 45 m/s) of the right median nerve. Sensory nerve conduction studies could not be performed due to an increased muscle tone. Electromyography of the rectus femoris, deltoid, and biceps brachii muscles showed pathological spontaneous activity (PSA) in the form of positive sharp waves (PSWs), fibrillations, and fasciculations, as well as myokymia, but no myotonic discharges. In summary, the electrophysiological findings were compatible with a lower motor neuron pathology. Again, no anti-neuronal antibodies were found in the serum. Unfortunately, the patient neither consented to a lumbar puncture nor to any other diagnostic or therapeutic attempts including plasma exchange. She decided to discontinue her hospital stay and left against medical advice.

## 3. Results

While in 2012 Cerami and colleagues reported the first case of acquired NMT following qHPV vaccination [6] and in 2013 Sedarous and Lange reported the first case of NMT with demyelinating CNS lesions [7], this is the first case where both aspects occur in the same patient. However, the case of Sedarous and Lange was an undefined, probably genetic disease including NMT as a symptom, but likely not of autoimmune origin. The clinical courses of the patients in the aforementioned reports share key aspects with ours, such as well-described NMT symptoms like muscle twitching and cramps, impaired muscle relaxation, and hyperhidrosis. However, a unique clinical feature of our patient was the non-fixed swan neck deformities of the fingers, which is rare in NMT without an underlying rheumatologic disease, of which we did not find any evidence [8]. Pes cavus, on the other hand, has been described in certain genetic syndromes featuring NMT. Accordingly, it would have been interesting to investigate our patient for rare genetic disorders which was unfortunately prevented by her reluctance to further diagnostic measures. Paraclinically, we found neither neuromyotonic discharges in myography nor positive antibodies against CASPR2 and LGI1 in the serum. Of note, their absence does not exclude the diagnosis of acquired NMT, as they are only detected in about 40–50% of cases [9]. However, the most intriguing aspect of this case is the presence of partly active brain lesions and oligoclonal bands (OCBs), which are both typically seen in MS. In this regard, our patient formally fulfilled the 2017 McDonald diagnostic criteria of dissemination in time (presence of Gad-enhancing and -nonenhancing lesions, positive OCBs) and space (periventricular, juxtacortical, and spinal lesions). Even without typical clinical symptoms of MS relapses (e.g., optic neuritis), it is therefore tempting to diagnose our patient with both radiologically isolated syndrome (RIS) and NMT. Even more so, it is conceivable that NMT may mask MS symptoms. Moreover, as a young female, our patient was at a generally higher risk of developing inflammatory CNS disease. On the other hand, in some cases NMT has been reported to manifest itself in the CNS leading to symptoms such as hallucinations, chorea and insomnia, first described by 19th century French physician Augustin Morvan [10]. In addition, the aforementioned case of Sedarous and Lange also featured OCBs in the CSF and periventricular lesions even though there was no evidence of dissemination in time and space [7]. Their patient presented primarily with muscle twitching and cramps, hyperhidrosis, and memory disturbance. Moreover, he experienced bouts of positional vertigo with no clear central correlation. Apart from “poor memory” and the nonspecific vertigo, Sedarous and Lange did not find any clear evidence of CNS dysfunction and attributed the patient’s complaints to peripheral nervous system (PNS) hyperexcitability. In summary, based on her symptoms, our patient does not fall into the category of “classical” Morvan’s syndrome. Finally, vaccines have been described as a possible trigger factor for a small percentage of certain inflammatory autoimmune conditions affecting the CNS and PNS such as acute disseminated encephalomyelitis (ADEM) or Guillain-Barré Syndrome (GBS), respectively. Accordingly, Sutton and colleagues described five cases of clinically isolated syndrome/MS flare-ups following HPV-vaccination. They argued that this may be attributed to the stimulatory effects of HPV virus-like particles on the immune system via “bystander activation” [11]. Even though human papilloma viruses are known for their strict affinity to epithelial cells [12], it is conceivable that the immunoregulatory signal provided for dendritic cells via IL-6- and TNF-α-secretion may also elicit an inflammatory response in the nervous system [13]. However, it is important to underscore that several studies in both genders have shown no significantly increased risk of autoimmune diseases following HPV-vaccination [14]. NMT *per se* is a rare disease and it is even rarer occurring after qHPV vaccination. In contrast, based on epidemiological data, inflammatory CNS diseases are comparatively frequent and mostly occur at a young age. Accordingly, there is a certain overlap between this group and people that receive HPV vaccination for the first time. This, in turn, could argue for coincidence rather than causality. Moreover, this fact could be the reason, why a comprehensive overview of case reports with NMT, MS, and patients with similar symptoms following qHPV vaccination is still unavailable. Nonetheless, awareness of rare but severe putative CNS and PNS complications of the qHPV-vaccine can help to better diagnose patients and will certainly help in the future to thoroughly analyze possible confounders in these cases.

## 4. Summary

HPV-vaccination is efficient in preventing HPV infection which is the greatest risk factor of cervical cancer. Quadrivalent HPV vaccination has been available for more than 15 years and has mostly no serious side-effects. However, the case presented here emphasizes the potential (auto)immunogenic properties of its viral components for both the PNS and CNS. Since our patient’s symptoms improved partially after the first vaccination dose but then recurred after the second, it is important to discuss the recent decision of the World Health Organization Strategic Advisory Group of Experts on Immunization (SAGE). This text states that based on the available data, protection against HPV from a single vaccine dose is considered to be comparable to that of two doses [15]. Of note, our patient had an unremarkable past medical history at the time of vaccination. However, in patients with pre-existing autoimmune conditions it might therefore be wise to administer only one vaccine dose.

## 5. Conclusions

For the clinician it is important to bear in mind that neuromyotonic symptoms in young qHPV-vaccinated patients might indicate inflammatory activity affecting both the PNS and CNS. Therefore, particular attention should be paid to symptoms occurring in the timeframe of three weeks to two months after antigen exposure where a high (auto)immunogenic vulnerability can be expected. Finally, irrespective of these considerations we would like to underline that vaccinations are an established, time-proven, and powerful tool to prevent disease. Their benefits far outweigh their potential disadvantages.

## Figures and Tables

**Figure 1 vaccines-10-01132-f001:**
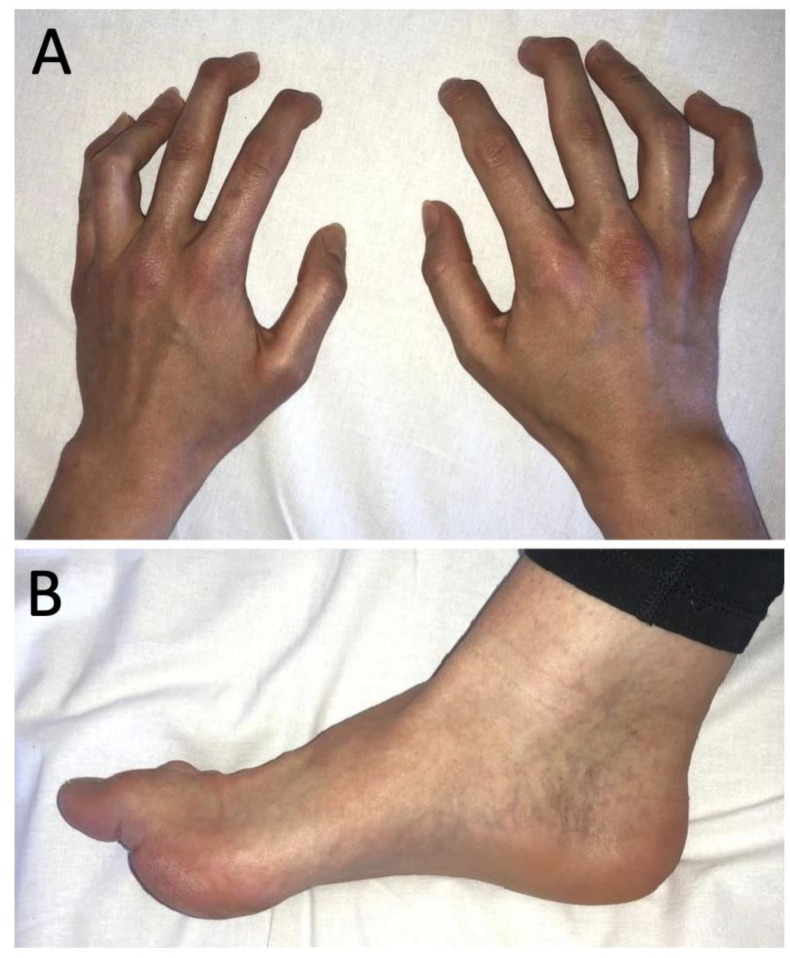
Clinical findings nine years after first manifestation of neuromyotonia. (**A**) Bilateral swan-neck deformities of fingers. (**B**) Bilateral pes cavus.

**Figure 2 vaccines-10-01132-f002:**
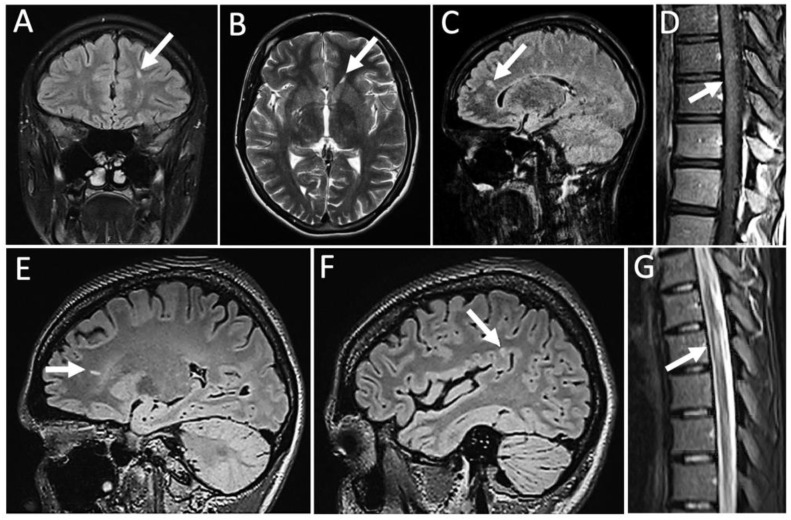
MR-imaging from 2013 (**A**–**D**) and 2021 (**E**–**G**), respectively. Sagittal (**C**,**E**,**F**) and coronar (**A**) FLAIR-sequences, horizontal T2w-sequence (**B**) and thoracolumbar sagittal sequences (**D**,**G**) with arrows indicating periventricular (**B**,**C**,**E**) and juxtacortical (**A**,**F**) lesions. Spinal lesions showing mild Gadolinium enhancement (**D**).

## Data Availability

Not applicable.

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
