# Peer review of "Neuromyotonia with Central Nervous System Lesions following Quadrivalent Human Papilloma Virus Vaccination"

_vaccines, 2022, doi:10.3390/vaccines10071132_

Round 1

Reviewer 1 Report

Comments follow throughout the attached document.

Author Response

We would like to thank the reviewers for their diligence during the revision of this manuscript. We feel that their remarks have improved this manuscript significantly and hope that the revised version will meet their expectations.

Reviewer 2 Report

The case report by Hatami and coauthors presents a rare case of NMT possibly due to HPV.

There are number of issues that needs to be addressed in this study:

1.       I don’t think that Germany has no regulation for ethical approvals for this type of study. When you are reporting any study involving living system, ethical approval is required.  In case, if it is not required who regulated their written informed consent from patients.

2.       Very strange that the patient has taken two doses within a span of two to three months. As per CDC, there should be a space of 6-12 months https://www.cdc.gov/vaccines/vpd/hpv/public/index.html

3.       The discussion section is just a summary, so it is better to rename it as summary.

4.       If interested, add a take home message in few lines as conclusion.

5.       In figure 2, all images need to be of similar resolution to understand the exact change that has happened over a period.

6.      HPV vaccination is one the important vaccination drive by WHO. Recent SAGE’s review concluded that a single-dose Human Papillomavirus (HPV) vaccine delivers solid protection against HPV, the virus that causes cervical cancer, that is comparable to 2-dose schedules. Could you please elaborate whether there are any special factor that has acted as triggers for NMT in this patient? Also, one dose schedule further decreases the risk of this type of manifestations.

7.      More than 10 million doses of HPV was given in a small country like U.K upto 2018. There are no reports till date of NMT. It has been reported to reduce the chances of cervical cancer by 86%. I think, there is a need for possible confounders in these rare cases of NMT or earlier reports on autoimmune reactions due to HPV. I suggest authors can elaborate on it in his discussion.

8.      There are vaccine opposers all around the world, we need to be highly cautious while reporting any problem with the vaccines.

Author Response

(The authors gave the same response as above.)

Round 2

Reviewer 2 Report

Authors have appropriately addressed issues raised by me. I am convinced with their explanations and changes done in the manuscript. Take home message are statements that readers draw from the study and should be presented as 'Conclusion'.

Please change it as Conclusion.

The manuscript has improved significantly and it is publishable.

Author Response

We thank the reviewer for his suggestion and improved our manuscript accordingly.
